# Antioxidant, Antibacterial, and Antifungal Activities of the Ethanolic Extract Obtained from *Berberis vulgaris* Roots and Leaves

**DOI:** 10.3390/molecules27186114

**Published:** 2022-09-19

**Authors:** Khaled Meghawry El-Zahar, Mubarak Eid Al-Jamaan, Faisal Rasmi Al-Mutairi, Abdallah Mohamed Al-Hudiab, Mohamed Saleh Al-Einzi, Ahmed Abdel-Zaher Mohamed

**Affiliations:** 1Department of Food Science and Human Nutrition, College of Agriculture and Veterinary Medicine, Qassim University, Buraydah 51452, Saudi Arabia; 2Food Science Department, Faculty of Agriculture, Zagazig University, 44511 Zagazig, Egypt; 3College of Science and Humanities at Shaqra, Shaqra University, Shaqra 11961, Saudi Arabia

**Keywords:** *Berberis vulgaris*, antimicrobial activity, phytochemicals, food borne bacteria, phytopathogenic fungi

## Abstract

This work assessed the phenolic and flavonoid components and their antioxidant, antifungal, and antibacterial effects in the ethanolic extract of barberry leaf and roots. The antibactericidal activity of root and leaf extracts against pathogenic bacteria was tested using agar diffusion and microdilution broth production for the lowest inhibitory concentration (MIC). *Berberis vulgaris* root and leaf extracts inhibited *Staphylococcus aureus* ATCC9973, *Escherichia coli* HB101, *Staphylococcus enteritis*, and *Escherichia coli* Cip812. The disc assay technique was used to assess the bactericidal activity of the extracts versus both pathogenic Gram-positive and Gram-negative strains. Hydro alcoholic extract was more effective against bacterial than fungal strains. The results showed that *Berberis vulgaris* leaf and roots extract had similar antifungal activities. *Berberis vulgaris* root extract inhibited the mycelial growth of *Penicillium verrucosum, Fusarium proliferatum*, *Aspergillus ochraceous, Aspergillus niger*, and *Aspergillus flavus*. *Berberis vulgaris* root extract has excellent antioxidant, antibacterial, and antifungal effects. *Berberis vulgaris* exhibited antimicrobial activity in vitro, and MIC showed that *Berberis vulgaris* parts efficiently affected pathogens in vitro. In conclusion, both *Berberis vulgaris* roots and leaves have considerable antibacterial activity and can be used as a source of antibacterial, antioxidant, and bioactive compounds to benefit human health.

## 1. Introduction

Plant extracts have revealed several biological activities, which show antiviral, antibacterial, and antifungal properties [1]. Medicinal plants, which contain a wide range of phytochemicals, are among the most easily obtainable sources of such compounds [2]. Herbs are amongst the most effective objects to accomplish the goal. Antimicrobials must be used sparingly due to their hazardous effects. Due to the negative effects of standard antibiotics, herbal remedies are now recommended. Medicines contain herbs. Herb extracts or active components can be employed as additives in some foods [3]. There is a clear academic interest in therapeutic plants right now. Saponins, alkaloids, cardio protective flavonoids, pro-anthocyanidins, tannins, polysaccharides, fats, micronutrients, dietary fiber, phytate, phytic acids, and other phyto-components have been found in different parts of plants [4]. Barberry is considered an important solution to health problems when it comes to infectious diseases due to its high frequency [5]. Since disease resistance to antimicrobial chemicals is increasing, additional research is being conducted to discover novel antimicrobial drugs. A large range of medicinal activities and nutritionally important phytoconstituents has been identified in barberry [6]. *Berberis vulgaris* also contains antihypertensive, anti-inflammatory, antioxidant, anticancer, antimicrobial, and hepatoprotective properties that are used to cure different diseases [7]. *Berberis vulgaris* contains ascorbic acid, triterpenoids, vitamin K, more than 10 phenols, and much more than 30 times alkaloids. The major alkaloid isolated from *Berberis vulgaris* is berberine [8]. According to studies, it possesses antibacterial, antiparasitic, antihepatotoxic, anticancer, antitriglyceridemic, antidiabetic, and antioxidant properties. In the Middle East, Europe, and the United States, the plant’s roots, stems, and leaves are used for a number of medicinal uses, including the treatment of diseases and as a harsh tonic [9]. Furthermore, *Berberis vulgaris* root extract efficiently treats and prevents stone development in the renal system [10]. Because smaller peptide fragments can more easily access microbial membranes than larger-sized complete proteins, the inhibitory activity of protein fractions is studied against *E. coli* HB101 and *S. aureus* [11]. Released peptides may have a higher content of positively charged amino acid residues, hydrophobic amino acid residues, or some amino acid residues of particular importance for antimicrobial action [12]. Furthermore, extract of *Berberis vulgaris* root has antibacterial properties versus *S. aureus, E. coli, Bacillus subtilis*, *Pseudomonas aeruginosa*, and *Candida albicans* [13]. Berberine has anti-inflammatory properties and can help in treating illnesses of the central and peripheral nervous system [14]. *S. aureus* was the most sensitive to *Berberis baluchistanica* extract, with an inhibition zone of 23 mm for bark, 22 mm for leaves, and 20.21 mm for root extracts. *Berberis baluchistanica* root extract inhibited the growth of *A. flavus* with 83%, *A. niger* 80%, and *M**ucor mucedo* 73%. Bark extract inhibited *M. mucedo* 86%, *A. flavus* 70%, and *A. niger* 60%. The leaf extract showed a significant inhibition against *M. mucedo* (83%), *A. flavus* (73%), and *A. niger* with 72%, respectively [15]. *Fusarium verticillioides* and *F. proliferatum* can produce carcinogenic fumonisins, posing serious risks to human and animal health [16]. Foodborne pathogens cause many illnesses with major health and economic effects. *S. aureus, Listeria monocytogenes, Bacillus cereus, Clostridium botulinum, E. coli, Clostridium perfringens, Shigella* spp., *Salmonella* spp., *Vibrio* spp., and *Yersinia enterocolitica* are investigated and have had notable outbreaks [17]. Cardiac failure, hypertension, inflammatory disease, malignancy, pharmaceutical damage, reperfusion, and brain disorder all have the common element of oxidative stress (OS) as a contributing factor in their pathogenesis. Plant parts with a wide spectrum of phenols, micronutrients, terpenoids, and other bioactive molecules strong in antioxidant capacity [18]. Antioxidants can slow the progression of fibrosis by regulating free radical damage. Antioxidants are recognized as preventive molecules, which may decrease OS in humans. Several fruits contain antioxidants, which can scavenge free radicals by adding an electron that turns them into safe compounds [19]. All polyphenol groups (phenolic, flavonoid, and pro-anthocyanidins) meet the structural capacity of scavenging free radicals and have anti-inflammatory, antibacterial, and antifungal properties in food [20]. This research aimed to determine the antioxidant and antimicrobial properties of *Berberis vulgaris* roots and leaves for use in wound healing. In order to achieve this goal, the antioxidant and antimicrobial activities of *Berberis vulgaris* roots and leaves extracts were assessed in vitro. We also analysed *Berberis vulgaris* components by HPLC.

## 2. Results and Discussion

*Berberis vulgaris* has a lengthy history of ethno-pharmacological application in the treatment of a variety of illnesses. Antioxidants prolong shelf life, decease waste production, reduce nutritional loss, act as antibacterial agents, and expand the range of lipids that can be utilized in certain foods [21].

### 2.1. Yield Extract, Antioxidant Activity and Total Polyphenol Compounds of Berberis vulgaris Leaves and Roots

The total phenolic and flavonoid contents of the leaf and roots of Berberis vulgaris were calculated as mg gallic acid equivalent (GAE) g^−1^, and the antioxidant capacity of these components was evaluated using DPPH^•^, β-carotene, and ABTS^•+^ methods. Alkaloids, tannins, glycosides, saponins, flavonoids, steroids, and terpenoids were all identified in the fractions of the ethanolic extract of *Berberis vulgaris*. The extract yields were 14.7 to 18.7 g/100 g for *Berberis vulgaris* leaf and root extracts. These findings coincide with those of Aliakbarlu et al. [22], who discovered that the average extract yield of barberry leaves extract was 18.7%. Variation in extract yields is attributed to differences in the polarity of compounds present in plants; Herrera-Pool et al. [23] have reported such differences. *Berberis vulgaris* leaf extract has 120.7 mg GAE g^−1^, whereas roots have 147.2 mg GAE g^−1^ (Table 1). Flavonoids have a wide range of biological and chemical properties, including the ability to scavenge free radicals. As a result, the total phenolic and flavonoid amount of the extracts was determined. The results obtained in this investigation from *Berberis vulgaris* extracts are comparable to those found by Eroğlu et al. [24].

Table 1 shows that *Berberis vulgaris* leaf and roots contain 68.3% and 93.1% antioxidants, respectively. *Berberis vulgaris* extracts have equivalent antioxidant capacity to the BHA used in carotene bleaching (97.2%). Our findings were similar to the results of Motalleb et al. [25] who determined the antioxidant activity of *Berberis vulgaris* fruits by the β-carotene bleaching method as 73.62% in ethanol extract. DPPH^•^ is a strong oxidant that takes an additional electron or hydrogen radical in order to become a stable element [26]. OS present during the lipid peroxidation process is expected to play a crucial role in a variety of disorders, including cancer and cardiovascular disease [22,27]. The DPPH^•^ radical scavenging capabilities of the root and leaf extract of *Berberis vulgaris* are depicted in Table 1. The antioxidant activity of *Berberis vulgaris* root extract was approximately 44.3%, followed by leaf extract at 21.4%. The content of phenolic components in the extracts is primarily responsible for the antioxidant function of the extracts [28]. Extract-derived components can scavenge free radicals via electrons or hydrocarbons, preventing damaging radical chain interactions. Despite its limitations, a β-carotene /linoleic acid lipid–water emulsion test was used to evaluate the extracts [29]. In this experiment, linoleic acid oxidation produced hydroperoxide-derived free radicals that attack the chromophore of β-carotene, resulting in the bleaching of the reaction emulsion. As shown in Table 1, extracts can prevent the bleaching of β-carotene by scavenging free radicals produced from linoleate. It has been proposed that the polarity of an extract is significant in water–oil emulsions, as non-polar extracts are more effective antioxidants than polar extracts due to a concentrating effect within the lipid phase [29]. Thus, it would be expected that the less polar extracts would be more potent. This phenomenon was not observed in the case of extracts studied here, a finding which has also been reported previously [29]. Evidence from β-carotene/linoleic acid bleaching suggests that the extracts have the ability to scavenge free radicals in a highly diverse environment. It is possible that this means the extracts could be used as an antioxidant preservative in emulsions. In conclusion, we used ABTS^•+^ techniques to evaluate the antioxidant properties of *Berberis vulgaris* leaf and root extracts (Table 1). The antioxidant capacity of *Berberis vulgaris* was tested by Özgen et al. [30] who determined the ABTS^•+^ antioxidant capacity between 41.1 and 49.3 Trolox equivalent (TE) at mmol/L^−1^. The lowest and the highest ABTS^•+^ results were observed for leaves and roots extracts with 71.5% and 86.5%, respectively. Furthermore, the results demonstrate the extracts’ capability to remove OS in various systems, implying that treatment medicines targeting radical-related clinical damage could benefit from them. A positive relationship between the DPPH^•^ scavenging ability, ABTS^•+^, and β-carotene bleaching extent was confirmed. Generally, a positive correlation between TPC and antioxidant capacity was reported.

### 2.2. Phenolic and Flavonoid Compounds in Barberry Extracts Identified by High-Performance Liquid Chromatoghry (HPLC)

*Berberis vulgaris* extracts contain a high concentration of polyphenols. Comparison with reference compounds, analysis of scientific literature, and investigations on their mechanism of fragmentation led to the provisional identification of 23 different secondary metabolites in different plant organs. The highest concentration was discovered for derivatives of benzoic and cinnamic acids among the identified polyphenol-flavonoid components. Table 1 displays the complete list of detected metabolites in the examined samples, including simple organic acids and sugar acids from various organs of *Berberis vulgaris.* Simple phenolic compounds were detected in the chromatograms of *Berberis vulgaris* root samples. The roots had trace amounts of quercetin, vanillic acid, and rosmarinic acid in addition to caffeic acid, chlorogenic acid, myricetin, and catechin derivatives. The results of this study corroborated those of previous investigations into the alkaloid profile of this species [31]. HPLC chromatographic study shows that there is 0.6 mg of the active component berberine per 1 mg of *Berberis vulgaris* extract (Figure 1). After determining that 19 and 45 mg/mL of berberine could be detected and quantified, respectively, these values were kept as constants. Concerns about the potential side effects of synthetic medication have prompted a rise in the use of natural materials as an adjunct to conventional therapy for the restoration and treatment of various illnesses in recent years. Since the roots and leaves of the *Berberis vulgaris* plant exhibited significant antioxidant activity, we used high-performance liquid chromatography to analyses them for the presence and concentration of phenolic compounds. Gallic acid, berberine, chlorogenic acid, vanillic acid, caffeic acid, syringic acid, rutin, p-coumaric acid, catechin, myricetin, and kaempferol were found in high concentrations of 18.2, 45.5, 75.2, 2.2, 33.5, 3.9, 8.1, 2.14, 40.2, 12.4, and 23.14 mg/mL extract, respectively. The most prevalent phenolic components in *Berberis vulgaris* roots and leaf extracts were identified to be chlorogenic acid, berberine, catechin, and caffeic acid. Kaempferol, rosmarinic acid, myricetin, luteolin, and p-coumaric acid were found in low amounts (Table 2). Kaempferol was the most prevalent flavonoid in *Berberis vulgaris* roots extract, with 23 mg/mL, followed by luteolin (19.7 mg/mL), and myricetin (12.5 mg/mL). Overall, barberry root extract contained more phenolic and flavonoid components than barberry leaf extract. Previous research has found that the greatest quantities of gallic acid, catechin, chlorogenic acid, vanillic, caffeic, syringic, ferulic acid, rutin, and o-coumaric in Jilin’s fruits were 0.182 g kg^−1^, 0.640 g kg^−1^, 0.624 g kg^−1^, 0.044 g kg^−1^, 0.089 g kg^−1^, 0.049 g kg^−1^, 0.031 g kg^−1^, 0.073 g kg^−1^,and 0.051 g kg^−1^, respectively [32]. Root extracts are rich in berberine, chlorogenic acid, and other alkaloid derivatives. Ethanol extracts from the roots contained the most isoquinolines, with 14% berberine, 23% chlorogenic acid, and 12.4% catechin. Our results are slightly above the typical alkaloid level in *Berberis* spp. based on these and other studies [33].

Other researchers have proved that the total phenolic contents of the *Berberis vulgaris* fruit extract were 33.06 mg GAE/g [24]. In another investigation, *Berberis vulgaris* and *Berberis crataegina* extract had total phenolic content of 73.48 and 71.6 µg GAE/mg, respectively [34]. Many factors can influence the levels of phenols, including geographic location, ambient temperature conditions, season of production, soil type, processing, and storage situations [22].

### 2.3. Antioxidant Activity of Berberis vulgaris Extracts

Antioxidants applied in numerous industries, including food technology and health, are vital to the scientific community. Understanding the kinetics and operational mechanisms of processes involving several antioxidants might be facilitated by chemical and biological approaches. There is no preferred method for evaluating the antioxidant properties of foods or extracts, as several techniques can yield radically divergent results [35].

### 2.4. Disc Assay of Berberis vulgaris Extracts against Tested Bacteria Strains

Several methodologies have been introduced to evaluate the antimicrobial capacity of plant constituents. The antioxidant and antimicrobial activities of the beneficial components were tested to establish the feasibility of the method of generating them without changing their chemical composition or nature. *Berberis vulgaris* root and leaf extracts are shown to exhibit widespread antimicrobial activity. All bacterial strains examined were inhibited via the leaf and roots of *Berberis vulgaris* extracts (Figure 2). *Berberis vulgaris* extract’s MIC was 150 g/mL against *Streptococcus mutans* Cip103220T, *E. coli* HB101, *E. coli* Cip812, *B. cereus* Cip5262, and *S. aureus* ATCC9973. *Berberis vulgaris* root and leaf extracts were antimicrobial against most tested bacteria. *S. aureus’* inhibitory zone was 1.9 and 1.75 cm, *E. coli* HB101’s was 2.3 and 1.95 cm, *S. enteritis’s* was 1.5 and 1.35 cm, and *E. coli* Cip812’s was 2.0 and 1.7 cm (Figure 2). Increasing *Berberis vulgaris* ethanolic extracts increased antibacterial activity. Disk diffusion MIC and MBC were used to measure antibacterial sensitivity.

#### 2.4.1. Antibacterial Assays

Initial disc diffusion experiments (100 g/disk) showed inhibitory zones of 1.8–2.5 cm for *Berberis vulgaris* root extract against all Gram-positive and Gram-negative bacteria, whereas leaf extracts were passive against all Gram-negative bacteria at 1.5–1.85 cm. Figure 1 shows root and leaf extract antibacterial broth dilution findings. All *S. mutans* treatments had 150 g/mL MICs. These data confirm flavonoids’ antimicrobial properties. The authors in [36] suggest potential uses for these metabolites in oral health. These results resembled those of Abdel-Naime et al. [37] who examined the effect of an ethanol extract on *S. aureus*. *B. cereus* has the highest inhibitory zone of 2.59 cm and the smallest MIC and MBC, while *S. enteritis* has the least MIC and MBC. The high antibacterial activity we found in this herbal essential oil suggests its potential application as a natural food preservative. Bactericidal activity of *Berberis vulgaris* extracts was validated using agar diffusion and a minimum inhibitory concentration of 40 g/mL of any extract [18]. In terms of antioxidants, the findings revealed that the microorganisms studied had a considerable antibacterial effect. This shows that the extracts have significant antibacterial properties and could be applied as natural food preservatives. Belofsky et al. [38] found a significant increase inside the antibacterial property for pure substances. Our research suggests that the synergistic effect of extracts with both antibacterial and antioxidant compounds may enhance their pathogenic antimicrobial capabilities. Isolating and identifying chemicals associated with blocking the activity of harmful bacteria and studying the mechanism of this impact can be extremely advantageous.

#### 2.4.2. Turbidity-Based Bacterial Growth

*Berberis vulgaris* root and leaf extracts were investigated as possible antimicrobials (Figure 3). The chemical assessed in bacterial media has no influence on the growth of *B. cereus, S. enterica, Listeria innocna*, or *S. mutans* after four hours of incubation at 37 ± 1 °C. Root and leaf extracts of *Berberis vulgaris* inhibited *S. aureus, E. coli*, and *S. enteritis*. After one hour of incubation, an inhibitory effect occurred and persisted. The inhibitory efficacy of the substances evaluated did not alter considerably, though *E. coli* and *S. aureus* were especially affected. Regarding medication resistance and the intricacy of synthetic antibacterial ingredients, scientists are turning to natural sources such as plants’ antibacterial activity. *Berberis vulgaris* roots (Figure 3a) were the most bactericidal, followed by leaf extract (Figure 3b). Experiments maintain that *Berberis vulgaris* has antibacterial properties that previous researchers found [2,18]. The antibacterial activities of *Berberis vulgaris* extract may be attributable to berberine, although other constituents may also play a significant role. The mechanism of phenolic components’ toxicity against microbes involves the suppression of protein synthesis, RNA, and DNA, the instability of bacterial cell membrane structure, and the enhancement or diminution of enzyme activity [39,40].

### 2.5. Antifungal Assays

The antifungal activity was expressed as a minimum inhibitory concentration (MIC), as illustrated in Figure 4. The antifungal efficacy of *Berberis vulgaris* root and leaf extracts against seven phytopathogenic fungus strains is depicted in Figure 5. By increasing the amount of extract in the well, the exposed inhibition zone can be increased. *Berberis vulgaris* leaf and root extracts have comparable antifungal properties according to the findings. However, *Berberis vulgaris* root extract produced the highest levels of mycelial growth inhibition for *P. verrucosum, F. proliferatum, A. ochraceous, A. niger*, and *A. flavus*. For *P. verrucosum* and *A. ochraceous*, the maximum inhibition zones ranged from 1.7 to 2.35 cm at the 100 µL concentration. With an inhibitory zone ranging from 1.25 cm to 2.25 cm, *A. ochraceous* appeared to be more sensitive to all extract concentrations than other fungi, followed by *P. verrucosum* (1.0 to 1.5 cm), by increasing the volume from 50 µL to 100 µL. The inhibition zone produced by 100 L of extract was comparable to that produced by 50 µL of the antifungal drug DMSO 1 mg/mL, with inhibition zones ranging from 1.9 to 3.15 cm, respectively. *Berberis vulgaris* fractions inhibited the growth of several fungi [41,42]. According to Ghareeb et al. [21], a 62% berberine ethanolic extract from dried *Berberis vulgaris* roots shows potent antiviral and immune-modulatory properties. *Berberis vulgaris* extracts displayed antifungal activity against five fungal infections at dosages ranging from 1:1–1:8 (*P. verrucosum, F. proliferatum, Aspergillus parasiticus, A. niger*, and *A. flavus*). The screening antifungal activity of *Berberis vulgaris* roots and leaf extracts vs. pathogenic fungi, which exhibited significant antifungal action against *A. flavus, A. niger, P. verrucosum*, and *F. proliferatum*, was similar to the finding presented by Rehman et al. [41]. Because of its safety and lack of side effects, plant-based therapies have been widely employed to treat a variety of diseases [43]. Various researchers are currently scanning for great alternatives to decrease the health hazards associated with a natural formula of antibiotics [2,3]. Previous studies have been carried out on the chemical composition of *Berberis vulgaris* and have shown that the most important constituents of this plant are isoquinoline alkaloids such as berbamine, palmatine, and particularly berberine [44]. So far, various studies have demonstrated antibacterial and antiparasitic effects of this plant against several pathogenic strains due to having berberine [45]. Additionally, numerous studies have found that *Berberis vulgaris* and its major component, berberine, have antifungal action against *Candida* spp. [46]. Here, using an in vitro model, the antibacterial and antifungal properties of *Berberis vulgaris* extracts, particularly its active component berberine, were established (Figure 5). As mentioned above, based on the previous studies, the highest biological activity of the *Berberis vulgaris* was due to having isoquinoline alkaloids such as berberine. This alkaloid is sparingly soluble in water, and it is expected that the extraction contains negligible amounts of berberine, whereas ethanolic extract can draw this alkaloid from plant roots. In the case of antifungal effects of berberine, few studies approve the high potential of berberine against some pathogenic fungal strains [46,47].

The results suggest that each extract possesses a broad spectrum of antimicrobial potentials. The strong antibacterial effect of the obtained extracts is probably due to the presence of secondary metabolites, such as alkaloids, steroids, coumarin, saponins, and terpenoids, and a high content of phenolic and flavonoids which have been reported to be involved in the inhibition of nucleic acid biosynthesis and other metabolic processes [48]. Phytochemicals such as phenolics, flavonoids, saponins, and tannins are able to protect the protein membranes from denaturation by binding the cations and other molecules. According to Abate et al. [49], the higher the concentration of these classes of compounds, the lower the inhibition concentration IC_50_ needed to reduce the inflammation response. Therefore, polyphenolic compounds present in ethanolic extracts of bark leaves and roots could be the possible reason for the stabilization of the lysosomal membrane by its antidenaturation property.

## 3. Materials and Methods

### 3.1. Chemicals and Reagents

The following chemicals were obtained from Sigma-Aldrich: 2,2-diphenyl-1- picrylhydrazyl (DPPH), Gallic acid, quercetin, Dimethyl sulfoxide (DMSO), 3-(4,5-dimethythiazol-2-yl)-2,5-diphenyl tetrazolium bromide (MTT), and trypan blue dye (Steinheim, Germany). Merck provided trichloroacetic acid, disodium hydrogen phosphate (Na2HPO4), sodium dihydrogen phosphate (NaH_2_PO_4_), Folin–phenol Ciocalteu’s reagent, and sodium carbonate (Darmstadt, Germany). ABTS [2,2′-Azinobis (3-Ethylbenzothiazoline-6-Sulphonic Acid)], potassium persulfate, ascorbic acid, and BHT were obtained from Sigma-Aldrich, India. Analytical grade ethanol was purchased from Sigma-Aldrich (Las Vegas, CA, USA).

### 3.2. Plant Material

Barberry roots and leaves were purchased from the local market of Al-Qassim region as shown in Figure 6. The plant sample was identified and authenticated by the Staff of the Botany Department, College of Agriculture and Veterinary Medicine, Qassim University, KSA.

### 3.3. Preparation of Plant Extracts

The roots and leaves were chopped in a Waring blender and sifted through a wire screen (mesh size, 2 mm × 2 mm). The roots and leaves were then exhaustively extracted with 80% ethanol, 20% water (plant material solvent, 1:5). The resulting extracts were filtered and concentrated on a rotary evaporator under vacuum. The extracts were suspended in water containing 1% dimethyl sulfoxide (DMSO) to obtain a final 100 mg/mL concentration and then filtered through a sterilizing membrane with 0.22 μm [51].

### 3.4. Culture Media and Microorganisms

The antibacterial assay was conducted against five Gram-positive pathogenic bacteria: *Bacillus cereus* Cip5262, *Staphylococcus aureus* ATCC9973, *Listeria innocua* R1007, and *Staphylococcus enteritis;* and four Gram-negative bacteria: *Escherichia coli* HB101, *Escherichia coli* Cip812, *Streptococcus mutans* Cip103220T, and *Salmonella enterica* Cip5858. The strains were grown on nutrient agar broth at 37 °C for 24 hrs and kept in the refrigerator at 4 °C until subsequent use. The antifungal assay was carried out against seven fungal species, including *Aspergillus flavus* NRRL 3357, *Aspergillus niger*, *Aspergillus parasiticus, Aspergillus ochraceous, Aspergillus carbonarius, Fusarium proliferatum* MPVP328, and *Penicillium verrucosum* grown on potato dextrose agar (PDA) media. Microorganisms were obtained from the Institute of Microbiology, Bulgarian Academy of Sciences (CIMBSA, Sofia, Bulgaria) and the National School of Engineering Techniques of Agriculture and Food Industry (ENITIAA, Nantes, France).

#### Preparation of Standard Solutions

Gallic acid standard was prepared by dissolving gallic acid in 80% methanol; 7.5% Na_2_CO_3_, 5% NaNO_2_, 10% ACl_3_, 1 M NaOH solutions were prepared with double distilled water. Catechin and ascorbic acid standard solutions were also prepared with methanol.

### 3.5. Determination of Total Phenolic Compounds

The total phenolic compounds of *Berberis vulgaris* extracts were estimated using the Folin–Ciocalteu colorimetric method described by Škerget et al. [52]. The absorbance at 750 nm was measured to determine the total phenolic compound content. Gallic acid was used as the standard and expressed as mg of gallic acid equivalent (GAE/g^−1^) of barberry.

### 3.6. Determination of Total Flavonoids

The flavonoid content in barberry ethanol extracts was estimated by the AlCl_3_ method as described by Djeridane [53]. The absorbance of the reaction mixture was read at 430 nm with a spectrophotometer. Total flavonoid content expressed as quercetin equivalent (QE) was calculated using the following equation based on the calibration curve:Y = 0.0144 × −0.0092     R² = 0.9985(1)

### 3.7. High-Performance Liquid Chromatography Analysis

For the HPLC separation, a previously reported method was used for the determination of phenolic compounds with little modification [54]. A Waters 2695 Alliance liquid chromatograph (USA) equipped with a UV-Vis/DAD detector was used. Waters Sun fire C_18_ column (250 mm × 4.6 mm × 5 mm) chromatographic separation was performed. The separation was carried out with methanol and acetonitrile as a mobile phase at a flow rate of 1 mL/min. The column temperature was performed at room temperature (25 °C) throughout the experiment. Identification and quantification were carried out based on calibrations of the standards prepared from phenolic acids dissolved in a mobile phase. Retention time and peak area were used for calculation of phenolic acid and flavonoid compounds by the data analysis of Waters Software.

### 3.8. Determination of Total Antioxidant Capacity by the DPPH^•^, ABTS^•+^, and β-Carotene Methods

The DPPH^•^ free radical test was neutralized using a modified method of Esmaeili et al. [55]; 1 mL of 0.1 mM solution of DPPH^•^ in methanol was mixed with 2 mL of the *Berberis vulgaris* extracts at different concentrations. The mixture was then incubated at room temperature for 30 min in the dark. The control was prepared by mixing 1 mL of DPPH^•^ solution with double distilled water. The absorbance was measured against a blank at 517 nm using spectrophotometer (Systronics Visiscan 167). Lower absorbance of the reaction mixture indicates higher DPPH^•^ free radical scavenging activity. Ascorbic acid was used as the standard. The percentage of antioxidant activity of free radical DPPH^•^ was calculated as follows:Inhibition (%) = [(A_control_ − A_sample_)/A_control_] × 100(2)

The samples’ radical scavenging activity against ABTS^•+^ radicals was tested using the adapted method of Shah and Modi [56]. The ABTS^●+^ stock solution was prepared by reacting ABTS aqueous solution (7 mM) with 2.45 mM aqueous solution of potassium persulfate in equal quantities; the mixture was allowed to stand in the dark at room temperature for 12–16 h before use. The working solution of ABTS^●+^ was obtained by diluting the stock solution in methanol to give an absorbance of 0.70 ± 0.02 at 734 nm. Then, 2.0 mL of ABTS^●+^ solution was mixed with 1 mL of the *Berberis vulgaris* extracts at different concentrations. The mixture was then incubated at room temperature for exactly 10 min in the dark. The control was prepared by mixing 2.0 mL of ABTS^●+^ solution with 1 mL of double distilled water. The absorbance was measured against a blank at 734 nm using a spectrophotometer (Systronics Visiscan 167). BHT was used as the standard. Samples were prepared and measured in triplicates. The percentage of scavenging activity of each extract on ABTS^●+^ was calculated as % inhibition (I%) using the following equation:I% = [(Ao − As)/Ao] × 100(3)
where A_o_ is the absorption of control, and A_s_ is the absorption of the tested extract solution. The antioxidant activity (%) of *Berberis vulgaris* extracts was evaluated in terms of the bleaching of the β-carotene relating to BHT [57]. The results were expressed as a BHT-related percentage. A 15 µg/mL standard stock solution of β-carotene and linoleic acid was prepared by dissolving 0.5 mg of b-carotene in 1 mL of chloroform and adding 40 mg of linoleic acid together with 400 mg of Tween 40. The chloroform was evaporated. Then, 100 mL of aerated water was added to the residue. Reference compound (BHT) and sample extracts were prepared in methanol. The emulsion (3 mL) was added to a tube containing 0.2 mL of different concentrations of extract and essential oils (500, 700 and 1000 µg/mL). The absorbance was immediately measured at 470 nm, and the test emulsion was incubated in a water bath at 50 °C.
Antioxidant activity (%) = [1 − (A^0^_sample_ − A^120^_sample_)/ (A^0^_control_ − A^120^_control_)] ×100(4)

### 3.9. Antibacterial Activity

Antibacterial activity was determined by monitoring the inhibition zone in the solid media around the injected material and tracking bacterial growth using visual turbidity in the liquid media [58].

### 3.10. Liquid Media Technique

Using the Brain Heart Infusion (BHI) media, a preculture of every examined bacterial strain was generated for 24 h. A portion of 50 μL of preculture was combined with 50 mL of sterilized BHI media in a 250 mL Erlenmeyer flask. The mixture was continually stirred at the appropriate temperature of the examined bacterial strains until the optical absorbance of the media reached 0.5 at 600 nm. Then, 4 mL of the resulting suspension was added to each tube with 80 mL of the tested compound (5 mg/mL) for a final substance concentration of 100 g/mL. The tubes were kept at the optimum temperature for every investigated strain for 0, 1, 2, 3 and 4 h, respectively. The optical absorption at 600 nm was measured and taken as a metric of bacterial growth, with a decrease in absorption indicating bacterial suppression.

### 3.11. Solid-Media Techniques (Mass Diffusion)

A portion of 10 μL from each strain’s 24 h-BHI preculture was diluted with 10 mL of distilled water. The resulting solution was combined with 90 mL of gelose medium kept at 55 °C. The bacterium-containing medium was placed into petri dishes and allowed to harden for 30 min at 45 °C. Using a sterilized suitable tool, 5 mm diameter holes were induced in the solidified medium. The amount of the tested substances (80 μL adjusted at pH 5.5) was poured into each hole. After 10 min of diffusion at room temperature, the petri dishes were incubated at 37 °C for 24 h. The area of inhibition zone around the applied material was taken as an indication of the antibacterial activity. A blank sample was exactly prepared as the treatment exception in which the extract was substituted with soluble solvent dimethyl sulfoxide (DMSO). Positive controls were made as treatments except for gallic acid, retail, and synthesized antibiotics (tetracycline).

### 3.12. Antifungal Activity

Berberis vulgaris leaf and root extracts were investigated for antifungal efficacy against the following fungi in this study: *Penicillium verrucosum*, *Aspergillus flavus*, *Aspergillus niger*, *Aspergillus parasiticus*, *Aspergillus ochraceous*, *Aspergillus carbonarius*, and *Fusarium proliferatum*.

#### Growth Mycelial Inhibition

Aliquots of barberry extracts (100 mg/mL) and a positive (fungicide) or negative (1% DMSO) control were uniformly dispensed on petri dishes with potato dextrose agar (PDA) using a drigalsky handle. A mycelial disc (8 mm diameter) taken from the edge of a growing colony was placed in the center of each plate. The plates were incubated at 28 °C/2–8 days for the studied fungal strains. The positive controls were glyphosate (0.3–10 mg/mL), while the negative controls were DMSO (1%). Daily growth diameters were measured in four directions until the negative control colony reached the petri plate’s corners. The percentage of inhibition of the diameter growth (PIDG) of the extracts on fungal growth was determined according to the following formula [59]:PIDG = {[(Ed/Cd) × 100] − 100}(5)

Ed means the mycelia’s average diameter (mm) with the extractor fungicide, and Cd means the average diameter (mm) with the negative control.

### 3.13. Statistical Analysis

The experiments were conducted in triplicate, and results are expressed as mean ± standard error of mean. The data were analyzed using SPSS ver.22 (SPSS Inc., Chicago, IL, USA) by analysis of variance (one-way analysis *f* variance) and post hoc Tukey statistical test. Differences at *p* ≤ 0.05 were considered significant.

## 4. Conclusions

*Berberis vulgaris* extracts manifested variable amounts of antioxidant, antifungal, and antibacterial properties in a dose-dependent approach in several assaying methods. The antioxidant-capable extracts were shown to be natural equivalents to those with higher polyphenol contents. Therefore, extracts having a higher phenolic concentration proved to be more effective radical scavengers than extracts with a lower phenolic content. *Berberis vulgaris* leaf and root extracts contained effective compounds that could be used successfully as antibacterial and antifungal agents against a variety of microorganisms. *Berberis vulgaris* extracts have proven effectiveness in protection against human multi-resistant pathogens. This study confirms the various useful characteristics and features of *Berberis vulgaris* at a molecular level that can be used as a sustainable source for their potential nutritional applications for making functional foods in different food industries. However, further studies are required to elucidate the exact effects of these extracts and berberine in animal models as well as volunteer human subjects as a new therapeutic agent.

## Figures and Tables

**Figure 1 molecules-27-06114-f001:**
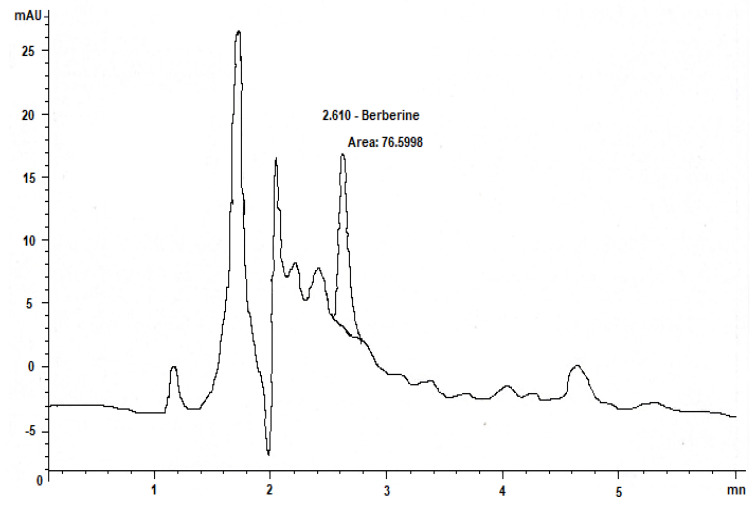
HPLC-chromatograms of polyphenolic compounds detected in *Berberis vulgaris* root and leaves.

**Figure 2 molecules-27-06114-f002:**
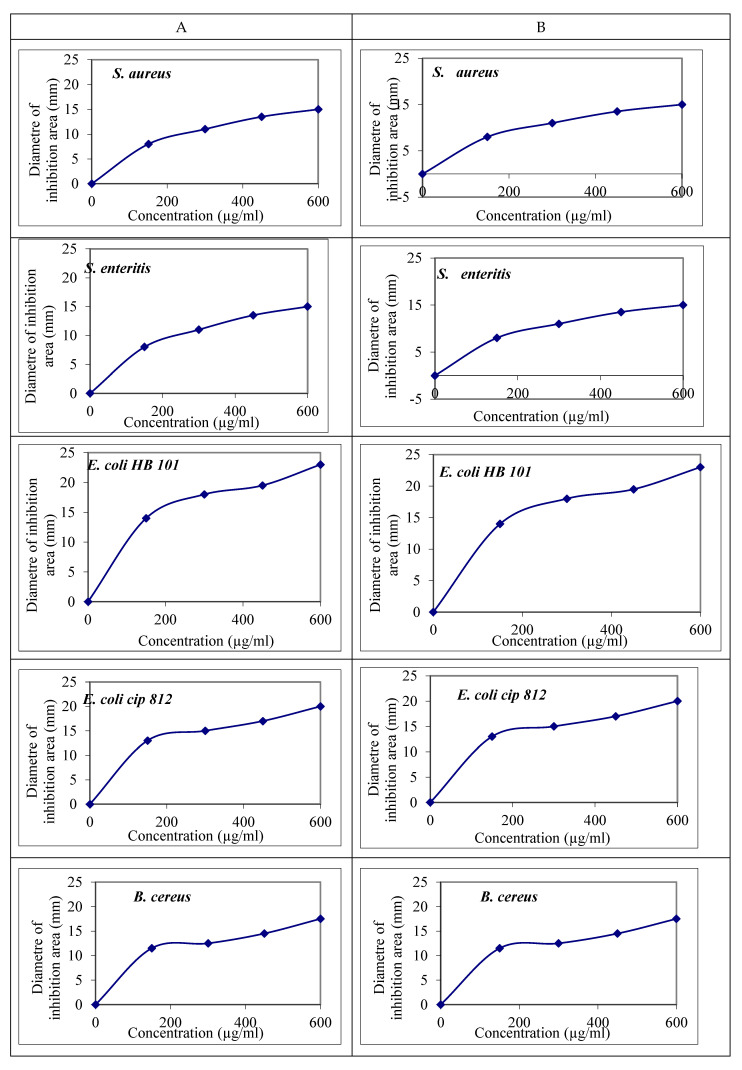
Relative antibacterial activity of *Berberis vulgaris* roots (**A**) and leaf (**B**) extracts against common food-borne pathogenic bacteria.

**Figure 3 molecules-27-06114-f003:**
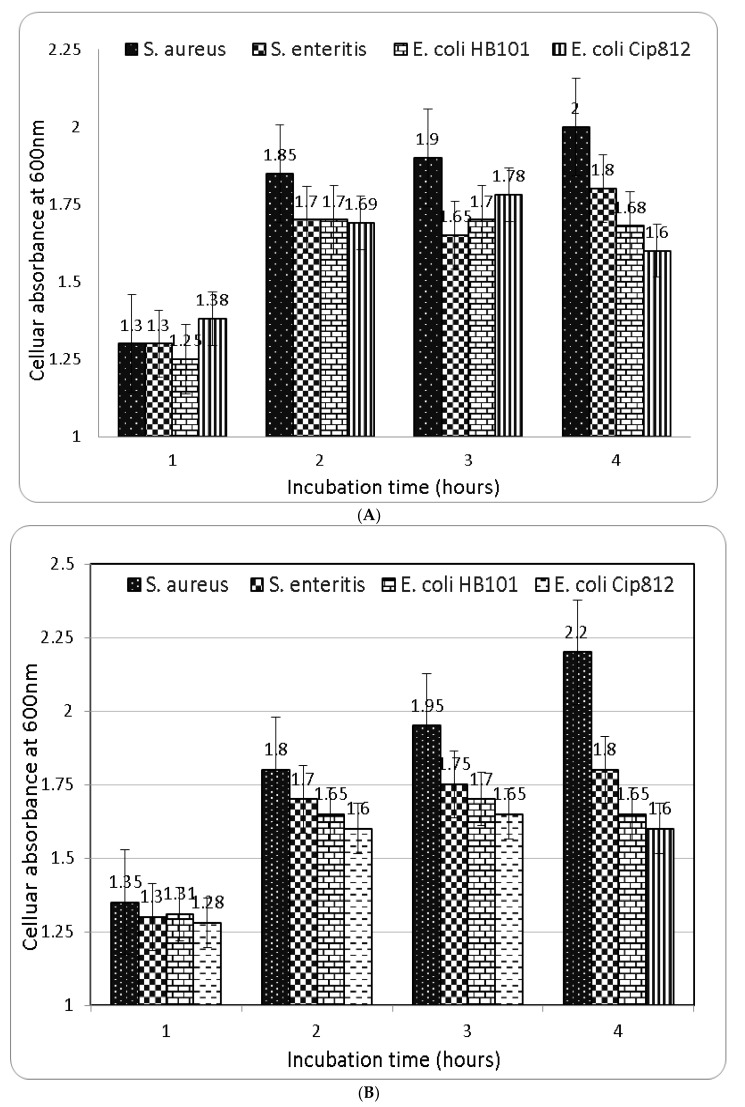
Antibacterial activity of *Berberis vulgaris* root (**A**) and leaf (**B**) against common food borne pathogenic bacteria.

**Figure 4 molecules-27-06114-f004:**
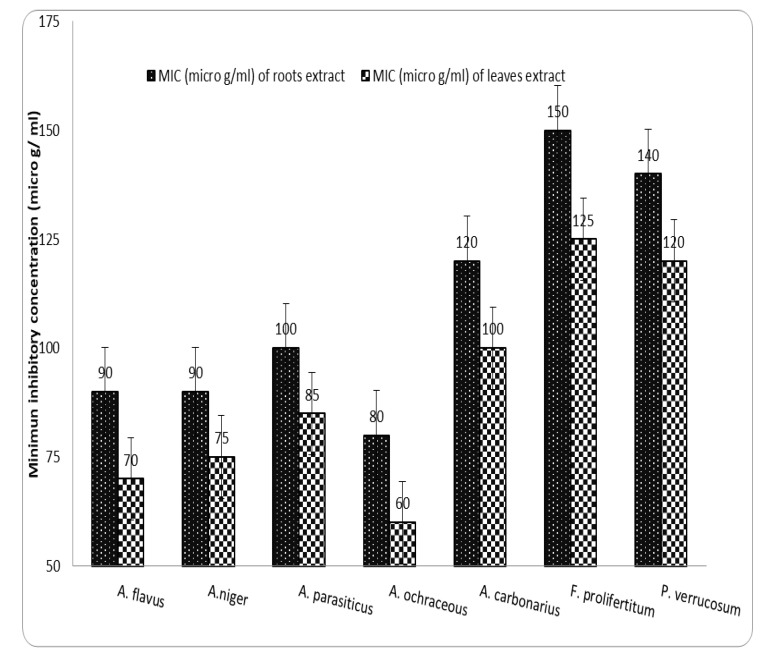
Minimum inhibitory concentration of *Berberis vulgaris* roots and leaf extracts for examined fungi.

**Figure 5 molecules-27-06114-f005:**
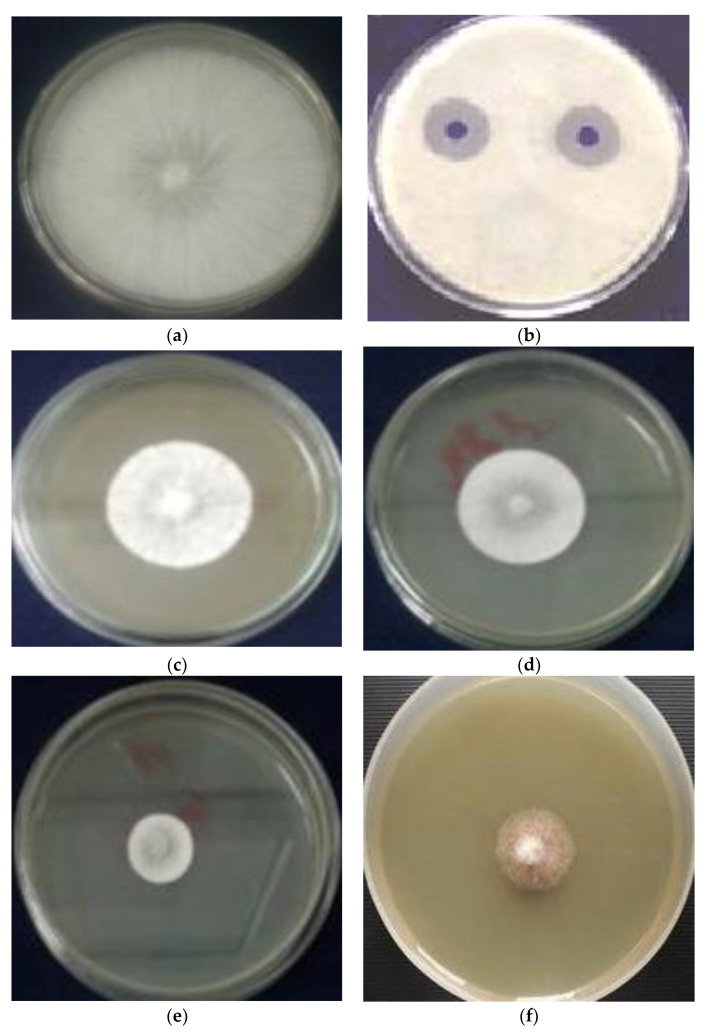
The effect of *Berberis vulgaris* extracts on fungal colony growth: (**a**) negative control; (**b**) positive control; (**c**) *P. verecossum*; (**d**) *A. parasiticus*; (**e**) *A. carbomareous;* and (**f**) *F. proliferatum*.

**Figure 6 molecules-27-06114-f006:**
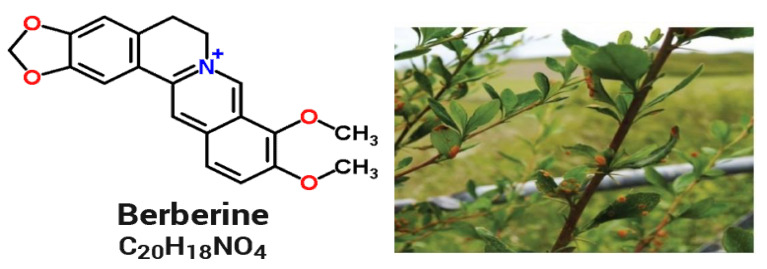
Berberine structure (**left**) and *Berberis vulgaris* tree (**right**) [50].

**Table 1 molecules-27-06114-t001:** Yield extract and polyphenol compounds in *Berberis vulgaris* leaf and roots extracts.

Phenolic and Flavonoid Compounds	*Berberis vulgaris* Ethanol Extract	
	Leaf	Roots
Total phenolic compounds (mg GAE/g^−1^ extract)	120.7 ± 1.2	147.2 ± 1.4
Total flavonoids (mg QE/g^−1^ extract)	59.58 ± 1.3	24.15 ± 0.8
Yield g/100 g DW	14.7 ± 1.5	18.68 ± 1.5
DPPH^•^	21.4 ± 0.2	44.3 ± 1.1
ABTS^•+^	76.5 ± 0.8	86.5 ± 0.8
β-carotene	79.50 ± 0.7	90.2 ± 0.9

**Table 2 molecules-27-06114-t002:** Phenolic and flavonoids concentration in *Berberis vulgaris* roots and leaves extracts as (mg/mL extract).

Polyphenolic Compound	Roots Extract	Leaf Extract
Resorcinol	3.96 ± 0.01	2.34 ± 0.01
Gallic acid	18.2 ± 0.15	10.2 ± 0.07
Catechin	40.2 ± 0.16	21.8 ± 0.17
Chlorogenic acid	75.2 ± 0.47	34.1 ± 0.22
Berberine	45.5 ± 0. 7	18.8 ± 0.2
Rosmarinic acid	20.2 ± 0.3	5.3 ± 0.17
Syringic acid	3.9 ± 0.001	3.2 ± 0.02
P-coumaric	2.14 ± 0.01	1.4 ± 0.01
Ferulic acid	2.9 ± 0.01	2.0 ± 0.01
O-coumaric	4.9 ± 0.03	4.2 ± 0.03
Prothocatechuic	2.9 ± 0.015	2.3 ± 0.01
Caffeic acid	33.5 ± 0.17	9.5 ± 0.06
Apigenin	8.06 ± 0.05	6.56 ± 0.03
Luteolin	19.68 ± 0.1	14.98 ± 0.06
Kaempferol	23.14 ± 0.1	20.44 ± 0.08
Rutin	8.1 ± 0.03	6.21 ± 0.02
Myricetin	12.4 ± 0.27	4.6 ± 0.11
Quercetin	1.1 ± 0.01	0.92 ± 0.006
Vanillic	2.2 ± 0.01	1.66 ± 0.01

Each value is a mean of three replicates ± SD.

## Data Availability

The authors state that all data and materials are available; they declare that the data is transparent for this manuscript.

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
