# Peer review of "Antioxidant, Antibacterial, and Antifungal Activities of the Ethanolic Extract Obtained from Berberis vulgaris Roots and Leaves"

_molecules, 2022, doi:10.3390/molecules27186114_

Round 1

Reviewer 1 Report (Previous Reviewer 3)

The manuscript has been modified and improved.

Author Response

Dear Sir,

Thank you for your useful comments and suggestions on the language and structure of our manuscript.

Reviewer 2 Report (New Reviewer)

the study is well designed and introduced. please use the same writing manar of microorganism names.

Author Response

Dear Sir,

Thank you for your useful comments and suggestions on the language and structure of our manuscript.

Comments

Responses

the study is well designed and introduced. please use the same writing manor of microorganism names.

Done: The microorganism’s names are reviewed in all the manuscript and written in italics, noting that the full name is written for the first time.

Reviewer 3 Report (New Reviewer)

The manuscript is well designed and experiments are properly conducted. Authors described the results satisfactorily and discussed it in the light of the current knowledge citing all relevant works. Conclusions are supported by the obtained results. However, before considering the MS for publication the following points/flaws need to be addressed;

- I propose a change of the title to: "Antioxidant, antibacterial and antifungal activities of the ethanolic extract obtained from roots and leaves of Berberis vulgaris"

-line 77: replace "role" by "factor"

-lines 57&58: "The inhibitory action of protein fractions is contrasted against E. coli HB101 and S. aureus". It's not clear what the authors mean by the word "contrasted". Please verify and replace by a clearer word to make the meaning better.

-Please abbreviate the word "oxidative stress: into "OS" throughout the MS, only written in full at the first mention.

-Why authors limited the study of the antimicrobial assays against bacterial (G+ and G-) and fungal pathogens?. Why not including a clinically-important yeast such as C. albicans?.

- Figure 2 legend is missing the word "against" between extracts and common.

-lines 368-369: Authors stated in in subheading 2.7. HPLC analysis the following "Identification and quantification were carried out based on 368 calibrations of the standards prepared from phenolic acids dissolved in a mobile phase". The chromatogram of the exact phenolic standards used and their retention times must be presented as supplementary material.

-For the zone inhibition assay, what standard antibacterial and antifungal drug were used?. This is very important control for comparative purposes. How were the inhibitory effects obtained with these standard antibiotics were compared to the ethanolic extracts?.

-Figure 4: label of the y-axis is missing.

-The discussion part lacks one paragraph detailing the proposed mechanism of action for the observed antibacterial and antifungal activities. I advise the authors to consult and cite the following papers:

Zouirech O, Alyousef AA, El Barnossi A, El Moussaoui A, Bourhia M, Salamatullah AM, Ouahmane L, Giesy JP, Aboul-Soud MAM, Lyoussi B, Derwich E. Phytochemical Analysis and Antioxidant, Antibacterial, and Antifungal Effects of Essential Oil of Black Caraway (Nigella sativa L.) Seeds against Drug-Resistant Clinically Pathogenic Microorganisms. Biomed Res Int. 2022 Jul 26;2022:5218950. doi: 10.1155/2022/5218950. PMID: 35958807; PMCID: PMC9363207.

Mssillou I, Agour A, Allali A, Saghrouchni H, Bourhia M, El Moussaoui A, Salamatullah AM, Alzahrani A, Aboul-Soud MAM, Giesy JP, Lyoussi B, Derwich E. Antioxidant, Antimicrobial, and Insecticidal Properties of a Chemically Characterized Essential Oil from the Leaves of Dittrichia viscosa L. Molecules. 2022 Mar 31;27(7):2282. doi: 10.3390/molecules27072282. PMID: 35408678; PMCID: PMC9000614.

Chebbac K, Ghneim HK, El Moussaoui A, Bourhia M, El Barnossi A, Benziane Ouaritini Z, Salamatullah AM, Alzahrani A, Aboul-Soud MAM, Giesy JP, Guemmouh R. Antioxidant and Antimicrobial Activities of Chemically-Characterized Essential Oil from Artemisia aragonensis Lam. against Drug-Resistant Microbes. Molecules. 2022 Feb 8;27(3):1136. doi: 10.3390/molecules27031136. PMID: 35164402; PMCID: PMC8840534.

El Abdali Y, Agour A, Allali A, Bourhia M, El Moussaoui A, Eloutassi N, Salamatullah AM, Alzahrani A, Ouahmane L, Aboul-Soud MAM, Giesy JP, Bouia A. Lavandula dentata L.: Phytochemical Analysis, Antioxidant, Antifungal and Insecticidal Activities of Its Essential Oil. Plants (Basel). 2022 Jan 25;11(3):311. doi: 10.3390/plants11030311. PMID: 35161292; PMCID: PMC8840530.

Author Response

Dear Sir,

Thank you for your useful comments and suggestions on the language and structure of our manuscript. 

Comments

Responses

line

I propose a change of the title to: "Antioxidant, antibacterial and antifungal activities of the ethanolic extract obtained from Berberis vulgaris roots and leaves"

Done

1

-line 77: replace "role" by "factor"

Done

78

-lines 57&58: "The inhibitory action of protein fractions is contrasted against E. coli HB101 and S. aureus". It's not clear what the authors mean by the word "contrasted". Please verify and replace by a clearer word to make the meaning better.

Done

58-60

-Please abbreviate the word "oxidative stress: into "OS" throughout the MS, only written in full at the first mention. 

Done

78,82,118,143

- Figure 2 legend is missing the word "against" between extracts and common.

-Figure 4: label of the y-axis is missing.                              

-Why authors limited the study of the antimicrobial assays against bacterial (G+ and G-) and fungal pathogens?. Why not including a clinically-important yeast such as C. albicans?.                   

We selected the bacterial strains and fungi studied in this research based on two factors. First, because we were able to obtain them through a joint cooperation with the Institute of Microbiology, Bulgarian Academy of Sciences (CIMBSA) and the National School of Engineering Techniques of Agricultural and Food Industries (ENITIAA), as we mentioned in the materials and methods section. The second reason is our inability to obtain any other strains, although there are many other clinically important fungal strains that can be used in the study.

-lines 368-369: Authors stated in in subheading 2.7. HPLC analysis the following "Identification and quantification were carried out based on 368 calibrations of the standards prepared from phenolic acids dissolved in a mobile phase". The chromatogram of the exact phenolic standards used and their retention times must be presented as supplementary material.

we upload a separate file as a supplementary material

-For the zone inhibition assay, what standard antibacterial and antifungal drug were used?. This is very important control for comparative purposes. How were the inhibitory effects obtained with these standard antibiotics were compared to the ethanolic extracts?

Done

446-448

-The discussion part lacks one paragraph detailing the proposed mechanism of action for the observed antibacterial and antifungal activities.

Done

304-315

This manuscript is a resubmission of an earlier submission. The following is a list of the peer review reports and author responses from that submission.

Round 1

Reviewer 1 Report

The manuscript "Antioxidant, antibacterial, and antifungal activities of Berberis vulgaris root and leaves ethanolic extract" has improved since the previous version.

1) Standard deviations could be added (with error bars) to the graphs.

2) Check the usage of italics in Material and Methods in the description of the bacterial strains used. In certain cases the italics are extended to the strain number, which is not correct.

3) Comments regarding figure 3 axis captions readability have been ignored. The figure need to be corrected.

Author Response

Dear Sir,

Thank you for your useful comments and suggestions on the language and structure of our manuscript. We have modified the manuscript accordingly, and detailed corrections are listed below point by point.

We have revised the whole manuscript carefully and tried to avoid any grammar or syntax errors. In addition, we have asked several colleagues who are skilled authors of English language papers to check their English. We believe that the language is now acceptable for the review process.

All figures and tables are provided and cited in sequence in the main text. We have checked all the references and formatted them strictly according to the Guide for Authors.

This table has been prepared to present the response to the reviewers' comments and the modifications made in the manuscript.

Comments

line

Responses

line

First Reviewer Comments

1) Standard deviations could be added (with error bars) to the graphs.

Done

Fig 3,4

271, 315

2) Check the usage of italics in Material and Methods in the description of the bacterial strains used. In certain cases, the italics are extended to the strain number, which is not correct.

Done

In all parts of the manuscript

--

3) Comments regarding figure 3 axis captions readability have been ignored. The figure needs to be corrected.

Done

271

Reviewer 2 Report

The paper has been only slightly improved in comparison with its first submission.    The work reported is still far from being considered for publication in Molecules. There are major problems with the phytochemical investigations reported in section 2.2. Phenolic and flavonoid compounds in barberry extracts identified by high-performance liquid chromatography (HPLC). I am still of the opinion the authors failed in carrying out an adequate method optimization. The active component berberine is not properly separated by the rest of the matrix (no matrix effects were investigated). They have just inserted the details concerning the HPLC analysis. No descrption of the stardards employed has been reported, as well as no information on the calibration curves employed.  Concerning the antoxidant activity ABTS data were provided withour proper comparison with phytochemical analysis. Overall, the novelty is still rather limited. English has been partially improved.

Author Response

Dear Sir,

Thank you for your useful comments and suggestions on the language and structure of our manuscript. We have modified the manuscript accordingly, and detailed corrections are listed below point by point.

We have revised the whole manuscript carefully and tried to avoid any grammar or syntax errors. In addition, we have asked several colleagues who are skilled authors of English language papers to check their English. We believe that the language is now acceptable for the review process.

All figures and tables are provided and cited in sequence in the main text. We have checked all the references and formatted them strictly according to the Guide for Authors.

Comments

line

Responses

line

Second Reviewer Comments

Overall, the novelty is still rather limited.

Done

25-29, 84-89

I am still of the opinion the authors failed in carrying out an adequate method optimization.

Done

153-166

The active component berberine is not properly separated by the rest of the matrix (no matrix effects were investigated).

Done

166-192

No description of the standards employed has been reported, as well as no information on the calibration curves employed. 

Done

366-376

Reviewer 3 Report

The authors improved the manuscript but are still some issues. In general, the authors do not demonstrate a careful writing of the text. There are errors due to lack of attention. Please reread the text word for word.

Major:

- Please include the methods for ABTS and b-carotene determination. There are not in manuscript.

- Please highlight in the text the originality of the manuscript.

Minor:

- L294 " Ethyl acetate": What did you use it for?

- Figure 3,4: please include standard deviation or other statistical descriptor.

Author Response

Dear Sir,

Thank you for your useful comments and suggestions on the language and structure of our manuscript. We have modified the manuscript accordingly, and detailed corrections are listed below point by point.

We have revised the whole manuscript carefully and tried to avoid any grammar or syntax errors. In addition, we have asked several colleagues who are skilled authors of English language papers to check their English. We believe that the language is now acceptable for the review process.

All figures and tables are provided and cited in sequence in the main text. We have checked all the references and formatted them strictly according to the Guide for Authors.

Comments

line

Responses

line

Third Reviewer Comments

- Please include the methods for ABTS and b-carotene determination. There are not in manuscript.

Done

384-390

- Please highlight in the text the originality of the manuscript.

Done

25-29, 84-89

- L294 " Ethyl acetate": What did you use it for?

Done

--

- Figure 3,4: please include standard deviation or other statistical descriptor.

Done

271, 315

Round 2

Reviewer 1 Report

Thanks for the corrections inserted. Please revise the X axis caption of Figure 3B.

Author Response

Dear Sir,

Thank you for your useful comments and suggestions on the structure of our manuscript. 

Comments

line

Responses

line

First Reviewer Comments

Thanks for the corrections inserted. Please revise the X axis caption of Figure 3B.

Done

Fig 3B

260

Reviewer 2 Report

The paper has no improvement compared to its previous submission. No remarks have been fixed (novelty, method optimization, description of standards and calibration curves). Also English (e.g. lines 27-29) is still terrible and the authors cannot be trusted when they claim they took the advantage of the help of skilled English speakers. 

Author Response

Dear Sir,

Thank you for your useful comments and suggestions on the language and structure of our manuscript. We have modified the manuscript accordingly, and detailed corrections are listed below point by point.

We have revised the whole manuscript carefully and tried to avoid any grammar or syntax errors. We believe that the language is now acceptable for the review process.

Comments

line

Responses

line

Second Reviewer Comments

No remarks have been fixed (novelty, method optimization, description of standards and calibration curves).

Done

65-71

107-109

295-301

346-349

376-382

385-398

402-409

Also English (e.g. lines 27-29) is still terrible

Done

26-28

Reviewer 3 Report

The manuscript has been modified and improved

Author Response

Dear Sir,

Thank you for your useful comments and suggestions on the language and structure of our manuscript.